# ECOACT: ECONOMIC AGENT DETERMINES WHEN TO REGISTER WHAT ACTION

## ABSTRACT

Recent advancements have enabled Large Language Models (LLMs) to function as *agents* that can perform actions[1] using external tools. This requires registering, i.e. integrating tool information into the LLM context prior to taking actions. Current methods indiscriminately incorporate all candidate tools into the agent's context and retain them across multiple reasoning steps. This process remains opaque to LLM agents and is not integrated into their reasoning procedures, leading to inefficiencies due to increased context length from irrelevant tools. To address this, we introduce `EcoAct`, a tool-using algorithm that allows LLMs to selectively register tools as needed, optimizing context use. By integrating the tool registration process into the reasoning procedure, `EcoAct` reduces computational costs by over 50% in multi-step reasoning tasks while maintaining performance, as demonstrated through extensive experiments. Moreover, it can be plugged into any reasoning pipeline with only minor modifications to the prompt, making it universally applicable to LLM agents now and in the future.

## 1 INTRODUCTION

Large language models (LLMs) have been conceptualized as agents and have demonstrated their capability to perform a broad range of complex tasks. When augmented with external tools (Yuan et al., 2023; Qu et al., 2024; Zhang et al.), LLM agents can extend their functionality beyond conventional natural language processing (Qin et al., 2023). For example, LLM agents equipped with scientific tools can conduct scientific research (Bran et al., 2023; Ghafarollahi & Buehler, 2024), while those integrated with physical robotic systems are capable of performing robotic manipulations (Ahn et al., 2022; Huang et al., 2023). External tools essentially expand the action space of LLM agents, enabling them to leverage existing functionalities to accomplish a variety of complex tasks (Xi et al., 2023; Wu et al., 2023a; Peng et al., 2023; Wu et al., 2023b; Shridhar et al., 2020).

To equip LLM agents with external tools, they must undergo a *tool registration* procedure. Specifically, information about the candidate tools needs to be added to the context of the LLMs that support the agents. This information represents essential details for tool usage, including tool names, descriptions in natural language, and instructions for input parameters. The current practice in tool registration indiscriminately incorporates all candidate tools into the agent's context, where these candidate tools are preemptively selected by users or retrieved automatically through external algorithms (Ocker et al., 2024; Qin et al., 2023; Gao et al., 2023). LLM-based agents will then process contextual information from all registered tools and select the appropriate tool for each reasoning step. However, this paradigm, which involves preparing all tools in advance and keeping the full information of the registered tools within the LLM's operational context, introduces one key issue: the tool registration process is opaque to the agents and not fully integrated into their autonomous reasoning pipelines. Each time the LLM is invoked, information from all passively registered tools is processed, even though not all tools are necessary and only one single tool can be utilized in each step, which drives inefficiencies in both cost and inference time (see Figure 1a). The problem becomes more pronounced as the number of pre-registered tools grows, imposing an even greater burden on the agent's decision-making process. The agent possesses the capacity to *reason to act* with their intrinsic reasoning mechanism but lacks the ability to *reason to register*.

---

[1] Unless otherwise stated, the term 'action' is defined as using a specific tool across the paper.

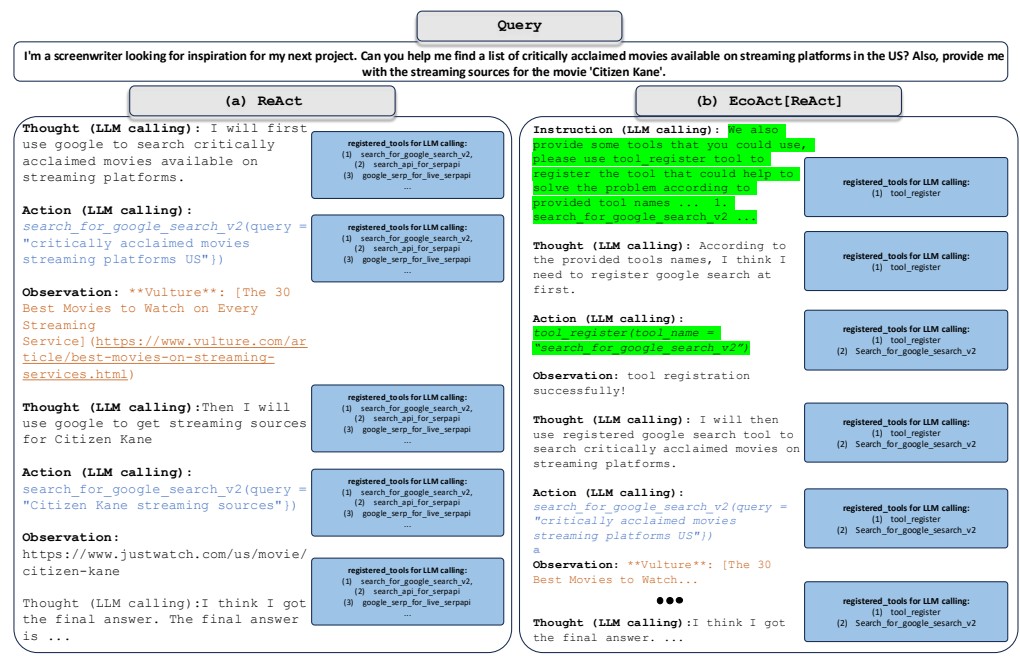

Figure 1: Overview of EcoAct, illustrating its effects after being integrated into the single-trace reasoning algorithm ReAct, which can serve as a fundamental component of complex reasoning methods. (a) In ReAct, all tools are registered in advance, retaining full information of these tools within the LLM's operational context at each reasoning step. This leads to unnecessarily long contexts, as tools irrelevant to the current problem remain included. (b) In contrast, EcoAct leverages ReAct's intrinsic reasoning capabilities to register only the tools deemed necessary, based on their concise and distinct identifiers - tool names, thus addressing the mentioned efficiency issues.

In this work, we present EcoAct, a general tool-using paradigm that integrates the tool registration procedure into the LLM agents' reasoning procedure, granting them discretionary authority, which is the freedom to register any tools they wish to use at any time through their intrinsic reasoning mechanisms (see Figure 1b). For any potentially useful tool, EcoAct prompts the agent to reason about registering the function before utilizing it, rather than passively accepting pre-prepared tools at each reasoning step. EcoAct gives agents the flexibility to register tools according to actual needs, thereby retaining only the necessary tools in the context and reducing costs. While the effectiveness of this tool registration process can be further enhanced through other agent reasoning methods (Yao et al., 2022; 2024; Qin et al., 2023; Wei et al., 2022), ensuring that tools are registered appropriately is essential for maximizing the agent's task-solving capabilities. Specifically, before the agent begins taking actions to solve the user's query using its intrinsic reasoning algorithms, we only provide the agent with one single meta-tool named *tool_register*, which enables the agent to register any tools deemed useful based on lightweight but easily-identifiable contextual information - tool names at any time step. The agent here will rely on this meta-tool to extend its skill library and solve the problem with its own self-registered tools. Additionally, the agent's intrinsic reasoning algorithms are seamlessly integrated into EcoAct. This integration enables the agent to employ its own reasoning logic to determine when and which actions to register.

We conduct extensive experiments on the ToolBench benchmark (Qin et al., 2023), which involves a diverse array of large-scale tools. We utilize EcoAct to enhance both the classic single-trace reasoning method ReAct (Yao et al., 2022) and the complex tree-structured reasoning method DFSDT (Qin et al., 2023), applying it across multiple models, The results show that the enhanced reasoning algorithms even can achieve monetary cost savings of over 50% on queries involving large tools from ToolBench, without compromising performance. Additionally, we conduct further analysis to demonstrate the effectiveness of key design choices in the proposed algorithm, regarding aspects such as the granularity of tool registration and the concise context used during tool registration.

Our contributions are summarized below: (1) We highlight a key limitation in the current tool-utilization paradigm of LLM agent systems: tool registration is essentially opaque to the LLM agents. Indiscriminately maintaining information about all registered tools within the LLM's operational context imposes a significant burden on the agent's decision-making process. (2) We introduce `EcoAct`, a plug-and-play algorithm that could seamlessly integrate tool registration into the agent's intrinsic reasoning procedures. The agent could *reason to determine when to register what tools* based on its needs, thereby mitigating the burden of processing all accessible tools in the backed LLM calling by only maintaining necessary tools. (3) We conduct comprehensive experiments using the ToolBench benchmark, which encompasses a wide range of large-scale tools. Our results demonstrate that the enhancement of `EcoAct` enables significant cost savings through various reasoning methods. Notably, for queries involving large tools from ToolBench, we observe cost reductions exceeding 50% across multiple models.

## 2 METHOD

In this section, we present `EcoAct`, a general tool-using algorithm designed to mitigate efficiency issues in agent tool-using scenarios. We begin by formulating the research problem and then provide the details of each component designed in `EcoAct`.

### 2.1 PROBLEM SETUP

We start by defining the relevant notations and outlining the research problem. Consider a language agent and a set of tools $\mathcal{Z} = \{z_i\}_{i=i}^{I}$ that the agent could access. The agent's objective is to address user queries according to a specific policy $\pi$. At any given decision time step $t$, the agent receives two types of information: **(1)** the historical context $c_t$ which includes all previous action-observation pairs, and **(2)** a set of accessible tools $\mathcal{Z}$ that can be used in this time step. The agent then must determine the next action to take. Formally, this decision process can be expressed as:

$$\pi(c_t, \tilde{\mathcal{Z}}) \to a_t, \ \text{ s.t. } \ a_t \in \mathcal{A}, \tag{1}$$

where $a_t$ denotes the action that been taken at time step $t$. It represents one specific tool-calling from accessible tool set $\tilde{\mathcal{Z}}$. $\mathcal{A}$ denotes the action space of this language agent. Consequently, the size of the tool space is equivalent to the size of the action space, i.e., $|\mathcal{A}| = |\mathcal{Z}|$.

In evaluating a specific agent algorithm, the total token consumption required to complete user queries, which encompass both input and output tokens—serves as a general metric for assessing the algorithm's cost (Chen et al., 2023; Wang et al., 2023; Hidvégi et al., 2024; Cheng et al., 2023). This is because token consumption is positively correlated with budget expenditure and latency, particularly in the context of large language models as a service (Gan et al., 2023; Sun et al., 2022). At time step $t$, we use the cost function $j(c_t, \tilde{\mathcal{Z}}, a_t)$ to represent the cost associated with making a decision at that step $t$. The one-step cost is given by:

$$j(c_t, \tilde{\mathcal{Z}}, a_t) = \alpha \cdot (l(c_t) + l(\tilde{\mathcal{Z}})) + \beta \cdot a_t, \tag{2}$$

where $l$ measures the token length. $\alpha$ and $\beta$ denote the cost per input token and output token, respectively, which are determined by the LLMs inference service provider. Under this formulation, the total cost $\mathcal{J}$ for completing users query with $n$ reasoning steps is:

$$\mathcal{J}^{\text{total}} = \sum_{t=1}^{n} j(c_t, \tilde{\mathcal{Z}}, a_t), \text{ where } a_t = \pi(c_t, \tilde{\mathcal{Z}}). \tag{3}$$

The focus of our research is to minimize the total cost $\mathcal{J}^{\text{total}}$ while maintaining a good performance in response to user queries.

### 2.2 ECOACT

**Motivation.** According to Equation 3, the polynomial $\mathcal{J}^{\text{total}}$ depends on $\tilde{\mathcal{Z}}$, $c_t$, and $a_t$. We primarily examines the token consumption associated with the input tool set $\tilde{\mathcal{Z}}$ at each time step, considering $c_t$ and $a_t$ as less controllable factors in practice. Most approaches for identifying $\tilde{\mathcal{Z}}$ at each time step

rely on a retrieval-based methods. A subset of tools is retrieved for each query and registered with the agent, which then makes sequential decisions until the problem is resolved Qin et al. (2023); Patil et al. (2023). However, a key limitation of this once-for-all paradigm is that each decision step processes contextual information from all retrieved tools, despite only one tool being utilized per step, which drives cost and latency. In Figure 2a, we present the ratio of tokens consumed by the necessary tools used at each decision step compared to the total number of tokens from all input tools, using the candidate tools retrieved by state-of-the-art tool using method AnyTool (Du et al., 2024) in ToolBench (Qin et al., 2023). It is observed that most portion of tokens is allocated to redundant tools rather than those that are actually executed. Except for cost issues, choosing from a large set of tools with extensive contextual information poses a challenge for large language models, as it results in a needle-in-a-haystack problem (Li et al., 2024). [2].

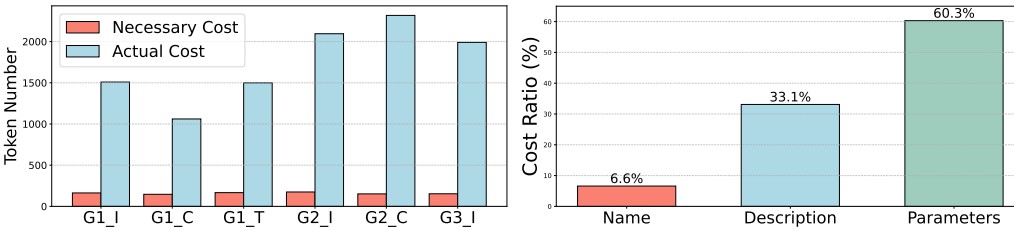

(a) Token cost: necessary vs. actual for tool-using     (b) Token cost ratios of different elements in tools

Figure 2: (a) Average token costs required for tools at each decision step, compared with the actual token costs incurred by tools using the React Algorithm (Yao et al., 2022), across six subsets of ToolBench (Qin et al., 2023). (b) Average token consumption percentages for each component of the tools in ToolBench (Qin et al., 2023).

**Overview.** The core concept behind `EcoAct` is to empower agents to autonomously register tools they find useful, rather than passively relying on pre-assigned tools. Two key questions arise in the design of this algorithm: (1) Is there a short yet distinct context that can assist agents in filtering and selecting the necessary tools from the available options without adding extra computational cost? (2) Given the answer to the first question, how can tool registration be seamlessly integrated into the agent's intrinsic reasoning process, enabling agents to determine, at each reasoning step, which tool to register and when, based on this distinctive information?

To address the first question, we propose utilizing tool names as easily recognizable tags to assist agents in determining which tools should be registered. To operationalize this, we introduce a meta-tool, *tool_register*, and define an action enabling agents to register any tools they consider relevant based on these tool names. The workflow is as follows: **(1) Initialization:** Prior to task execution, agents are equipped solely with the *tool_register* meta-tool. Simultaneously, users provide queries that include all available tool names along with instructions on how to use the tool register. Once the agent registers a tool by name using *tool_register*, detailed information about the registered tool becomes available. **(2) Reasoning:** At each reasoning step, the agent can either register a new tool or invoke a previously registered one, depending on the task requirements. We then detail the design of this process and the intuition behind each step.

**Tool name as informer.** The initial phase of `EcoAct` involves identifying a concise yet distinct context from the candidate tools, allowing agents to determine which tools are essential and which are not, utilizing their intrinsic reasoning capabilities without increasing computational load. As illustrated in Figure 2b, we present the average token consumption associated with each component of an external tool. We could observe that the majority of token consumption is attributable to tool descriptions and input parameter instructions, whereas tool names account for only 6.6% of the total token usage. Inspired by the efficiency of human tool using, wherein the utility of a tool can often be inferred from its name without recalling every specific detail, we propose leveraging tool names as the most easily identifiable markers to identify which tools require in-depth learning and which can be bypassed. Since tool names are often highly recognizable, this approach intuitively imposes minimal burden on the language model's context processing.

---

[2] https://github.com/gkamradt/LLMTest_NeedleInAHaystack

***tool_register* as meta-tool.** To achieve the objective of retaining only the necessary tools based on their names, we propose enabling agents to actively register tools using their intrinsic reasoning capabilities. Specifically, before the agent engages in a task, we (1) provide it with a list of tool names, which incurs minimal token usage, and (2) introduce a single tool, *tool_register*, which facilitates an action allowing the agent to register tools deemed useful based on their names at each time step. When the agent invokes *tool_register* with a selected tool name, it receives the complete information about the tool and adds it to its skill library. Essentially, we initialize the action space $\mathcal{A}$ with a single "meta-action," i.e., $\mathcal{A}_{t=0} = \tilde{a}$. Here, $\tilde{a}$ represents the action of using *tool_register* to register one tool, thereby expanding the action space over time. This strategy prevents the indiscriminate incorporation of all candidate tools into the context, fostering more efficient tool utilization.

Since tool registration has been integrated as a specialized action within the agent's action space, our algorithm offers several distinct advantages: (1) **Orthogonal to Agent Reasoning Algorithms:** Our method essentially forges a meta-tool capable of registering any tool deemed useful across various agent reasoning algorithms. As demonstrated in Section 3.3.1, it is agnostic to the specific reasoning algorithms employed and performs effectively with diverse reasoning techniques. (2) **Efficiency with Large-Scale Toolsets:** When dealing with queries with a vast number of tools, our method significantly reduces operational costs, achieving notable cost savings as detailed in Section 3.2. This efficiency arises because directly integrating a large number of tools into the agent is more costly. `EcoAct` minimizes the number of tools registered. Thus, our method is particularly beneficial in scenarios involving extensive toolsets, as it ensures that only the essential tools are being utilized. (3) **Simplicity and Intuitiveness:** Our method mirrors human problem-solving strategies involving multiple tools: it first filters tools based on simple identifiers (tool names) and then examines the details of the selected tools before use. This approach not only simplifies the process but also provides a general framework that could inspire the design of other agent algorithms.

## 3 EXPERIMENTS

We conduct experiments to prove the superiority of the proposed method. We begin by providing the experimental settings in Section 3.1. We then evaluate the `EcoAct` on ToolBench benchmark to verify its effectiveness in Section 3.2. Finally, we perform in-depth investigations in the last two sections to provide a better understanding of `EcoAct`.

### 3.1 EXPERIMENTAL SETUP

**Data preparation.** We mainly conduct experiments on the ToolBench (Qin et al., 2023), which is large-scale dataset for tool use. It involves 16,464 tools in total which has been widely used as the benchmark to make evaluations of tool use algorithm (Du et al., 2024; Ye et al.). ToolBench comprises six subsets G1-Instruction (G1-I), G1-Tool (G1-T), G1-Category (G1-C), G2-Instruction (G2-I), G2-Category (G2-C), and G3-Instruction (G3-I). These subsets are classified according to varying levels of complexity in tool use, with differences in 'Instruction', 'Category', and 'Tool' reflecting the relationships between tool categories in these test subsets and those in the training sets. Following the same setting with AnyTool (Du et al., 2024), we adopted the filtered benchmark which excludes all non-solvable queries in ToolBench. The remaining queries in these six subsets are 115, 132, 142, 107, 98, and 38, respectively. Unless specified otherwise, for each query, we use the state-of-the-art method AnyTool (Du et al., 2024) to retrieve tools for each query in all experiments across the paper. More details of this benchmark could be found in Appendix B.

**Evaluation metrics.** We primarily use two metrics to make evaluations: the pass rate and the cost, with the latter measured in monetary terms. Pass rate essentially measures LLM's ability to successfully execute an instruction within limited budgets. We utilize the evaluation script from (Du et al., 2024) to get the pass rate results in all experiments of our paper, which addresses issues related to artificially inflated pass rates (Du et al., 2024). We utilize GPT-4-turbo to make the pass rate evaluations, applying the same prompts as those used in ToolBench. Unless specified otherwise, we report the cost in US cents. More details about the evaluation prototype can be found in Appendix B.2.

Table 1: Comparison of the basic agent reasoning algorithm ReAct with its variant augmented with `EcoAct`. We show the pass rate performance and cost per query in US cents (¢) with different models in ToolBench (Qin et al., 2023) benchmark. We could observe that `EcoAct` significantly reduces costs associated with ReAct while maintaining comparable performance.

| Model | Method | G1 | | G2 | | G3 | | Average | |
|---|---|---|---|---|---|---|---|---|---|
| | | PR (%) | Cost (¢) | PR (%) | Cost (¢) | PR (%) | Cost (¢) | PR (%) | Cost (¢) |
| GPT-4-turbo | ReaAct | 16.2 | 6.3 | **18.5** | 8.1 | 13.2 | 11.5 | 16.0 | 8.6 |
| GPT-4-turbo | ReaAct **w/ EcoAct** | 16.7 | 5.9 (↓ 6.4%) | 18.0 | 6.1 (↓ 24.7%) | 13.2 | 7.7 (↓ 33.1%) | 16.0 | 6.6 |
| GPT-4o | ReaAct | 19.8 | 4.9 | 20.5 | 6.7 | 18.4 | 10.7 | 19.6 | 7.4 |
| GPT-4o | ReaAct **w/ EcoAct** | 20.1 | 3.7 (↓ 24.5%) | 20.8 | 4.8 (↓ 28.4%) | 21.1 | 5.8 (↓ 45.8%) | 20.7 | 4.8 |

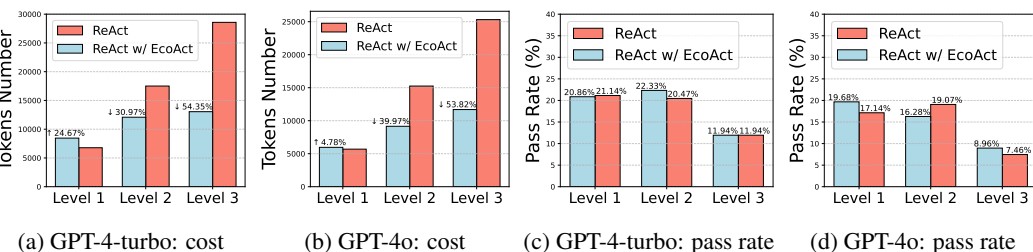

(a) GPT-4-turbo: cost     (b) GPT-4o: cost     (c) GPT-4-turbo: pass rate     (d) GPT-4o: pass rate

Figure 3: The average token cost and pass rate performance across queries with different numbers of tools in various models. For analysis, queries are categorized into three tool scale levels: Level 1, Level 2, and Level 3, corresponding to tool counts of 0-10, 10-20, and more than 20, respectively. It is observed that `EcoAct` benefits significantly from using a large number of tools, achieving token savings of 54.35% and 53.82% in two models respectively, with large-scale tools (Level 3). Additionally, `EcoAct` also surpasses the baseline on queries with large-scale tools in pass rate.

## 3.2 MAIN RESULTS

`EcoAct` essentially serves as a plug-and-play component for different agent reasoning algorithms. Here, we mainly evaluate the impact of `EcoAct` on the classical reasoning method ReAct (Yao et al., 2022). Our aim is to assess both its effect on the overall performance and its influence on the token cost associated with solving user queries. We mainly investigate our algorithm on ReAct because ReAct as the classic single trace agent reasoning algorithm could be regarded as the basic component of different agent reasoning algorithms (Yao et al., 2024; Qin et al., 2023) with multiple reasoning traces. We expect the conclusions drawn from ReAct experiments could potentially be generalized to more complex algorithms in the future.

**Overall results.** In this study, we compare ReAct with its variant augmented with `EcoAct` and present the aggregated results of different subsets in G1, G2, and G3 from ToolBench benchmark in Table 1, focusing on both performance and monetary costs, with different models. Our findings indicate that `EcoAct` significantly reduces the costs associated with ReAct while maintaining, and in some cases exceeding, its performance. This suggests that `EcoAct` serves as a "free lunch" component, enhancing ReAct without diminishing its reasoning capabilities. Given that ReAct functions as a foundational component for more complex reasoning algorithms, the experimental results highlight the potential for applying this cost-saving advantage to more sophisticated algorithms built upon ReAct. Interestingly, we observe minor performance improvements in some subsets, such as G1 on the GPT-4-turbo model and G3 on the GPT-4o model. These improvements may stem from our method's ability to address the "needle-in-a-haystack" problem (Li et al., 2024). By progressively expanding its tool library, the agent reduces the difficulty of selecting the most appropriate tool from a large set, thereby enhancing overall performance.

**Performance on multiple tool scales.** To better show the advantages of `EcoAct`, we also present the performance metrics and cost-saving percentages across various tool scales as assessed in the benchmark in Figure 3. We want to investigate the effect of our method for the queries with different tool scales. We could observe that `EcoAct` provides greater cost savings for queries involving large-scale tools, achieving token savings of 54.35% and 53.82%, respectively, with large-scale

tools (Level 3). This is due to the fact that traditional tools encounter higher costs when processing large-scale inputs, as they require the entire set of tools to be fed into the language model, resulting in increased expenses. In contrast, `EcoAct` addresses this issue by inputting only the complete information for registered tools, thereby avoiding redundant costs and optimizing overall efficiency. Additionally, we note a slight cost increase in Level 1 in some cases. This occurs because `EcoAct` introduces an additional LLM calling procedure for tool registration step. When the number of tools for specific queries is small, the cost of incorporating all tool information may be less than the cost of this extra LLM call, causing the advantages of our method to diminish.

## 3.3 MORE ANALYSIS

### 3.3.1 EXTENSION TO COMPLEX REASONING STRATEGY

Table 2: Comparison of multiple-traces reasoning strategy DFSDT (Qin et al., 2023) with its variant augmented with `EcoAct`. We could observe that `EcoAct` still could significantly reduces costs associated with DFSDT while maintaining comparable performance.

| Method | G2-instruction | | G3-instruction | |
|---|---|---|---|---|
| | PR (%) | Cost (¢) | PR (%) | Cost (¢) |
| DFSDT | 31.8 | 30.8 | **28.9** | 44.3 |
| DFSDT w/ `EcoAct` | **31.8** | **22.9 (↓ 25.7%)** | 26.3 | **21.8 (↓ 49.2%)** |

In this section, we examine the impact of `EcoAct` on the performance of the multiple-traces reasoning strategy DFSDT (Qin et al., 2023; Du et al., 2024). DFSDT allows agents to assess multiple reasoning paths and make informed decisions about whether to retract steps or continue along a promising path. The results, as shown in Table 2, indicate that integrating `EcoAct` with DFSDT results in notable cost savings while maintaining comparable performance on the most advanced model GPT-4o. Additionally, we observe that the cost per query in DFSDT is considerably higher than in the single-trace reasoning algorithm ReAct, for both our method and the baseline. This is due to the increased token usage and reasoning steps required by the multiple-traces approach. Consequently, the absolute cost savings achieved through our method are even more pronounced. These findings suggest that `EcoAct` is both versatile and beneficial across different reasoning methods. Whether applied to the single-trace reasoning of ReAct or the more complex DFSDT approach, `EcoAct` consistently enhances performance, affirming its effectiveness as a plug-and-play solution.

### 3.3.2 SKILL-LIBRARY EVOLUTION

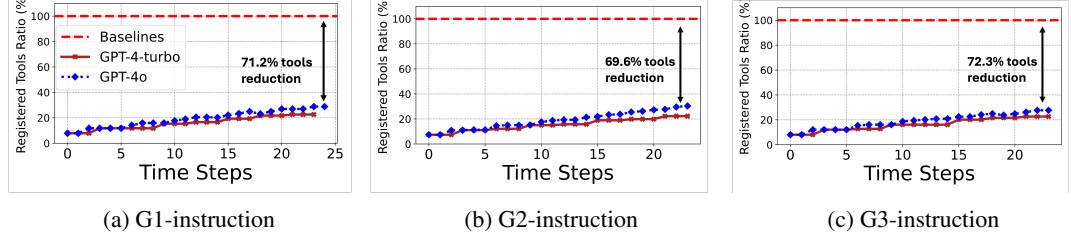

(a) G1-instruction      (b) G2-instruction      (c) G3-instruction

Figure 4: Evolution of the ratio of registered tools to total available tools across reasoning steps for different models, highlighting the largest percentage tool reductions for two models within each subset. Notably, the final registered tools comprise approximately 30% of the total available tools across all subsets, indicating that `EcoAct` effectively mitigates excessive tool registrations.

We then investigate how the number of registered tools evolves over time in the ReAct method when enhanced with `EcoAct`. Our primary objective is to investigate whether, in scenarios where a large number of tools are available for users' queries, `EcoAct` could cause ReAct to register an excessive number of tools greedily. Such behavior could lead the algorithm to revert to its original state by registering all available tools at the begining time, thereby undermining the benefits of `EcoAct`. To answer this question, we selected queries where the number of tools exceeded 20 from each group one of G1-I, G2-I, G3-I and conducted experiments to track the ratio of registered tools to the

total number of available tools over time. We also compared this ratio across different models. The results, averaged within each subset of data, are displayed in Figure 4.

From the results we could observe that (1) `EcoAct` is flexible. Tool registrations occur throughout the entire problem-solving process, suggesting that the agent is capable of registering any tool it deems useful at any point. Moreover, `EcoAct` may encompass a self-correction mechanism —if the agent realizes that a registered tool is unsuitable after obtaining more detailed information, it can leverage its intrinsic reasoning ability to register a more appropriate one in any time step. (2) We also could observe that the ultimately registered tools constitute only a small fraction of the total available tools, approximately 30% in all three subsets with all LLM models. The resukts demonstrates that, in cases where a large number of tools are available, most of them are redundant (about 70%), and `EcoAct` could effectively prevents the registration of these redundant tools.

## 3.4 ABLATIONS

### 3.4.1 SINGLE-TOOL VS. MULTIPLE-TOOL REGISTRATION

Table 3: We compared two tool registration mechanisms in `EcoAct`: (1) single-tool registration per step, and (2) multiple-tool registration per step. Experiments on G2/G3-I. subsets from ToolBench benchmark, using `EcoAct` to augment the ReAct algorithm on the GPT-4o model, revealed that multiple-tool registration led to a significant performance decline, even worse than standard ReAct.

| Method | G2-instruction | | G3-instruction | | Average | |
|---|---|---|---|---|---|---|
| | PR (%) | Cost (¢) | PR (%) | Cost (¢) | PR (%) | Cost (¢) |
| ReAct | 20.5 | 6.7 | 18.4 | 10.7 | 19.5 | 8.7 |
| ReAct w/ `EcoAct` (Single Tool Reg.) | **20.8** | 4.8 | **21.1** | 5.8 | **21.0** | 5.3 |
| ReAct w/ `EcoAct` (Multiple Tools Reg.) | 14.9 (↓ 5.9%) | **4.5** | 13.2 (↓ 7.9%) | **5.4** | 14.1 | **5.0** |

In our approach, the proposed meta-tool *tool_register* is designed to register only one tool per tool registration action in each time step. This naturally raises one critical question: could registering multiple tools simultaneously reduce costs while maintaining comparable performance? The intuition behind this is that each tool registration essentially require one LLM calling, which incurs token costs. If the agent could leverage its internal reasoning mechanisms to register several potentially useful tools in one interaction, it might lead to cost savings due to the decrease of LLM call number. To explore this hypothesis, we modified *tool_register* to allow for the registration of multiple tools at once, allowing agents to select as many tools as deemed necessary based on their reasoning. We conducted experiments using the G2-I and G3-I datasets in state-of-the-art GPT-4o model according to Table 1, where we use `Ecoct` to augment ReAct and present the results in Table 3.

From the results, we could observe that enabling *tool_register* to handle multiple tool registrations per action results in minor cost savings. However, the performance of `EcoAct` decreases significantly, with a 5.9% and 7.9% drop in the G2-instruction and G3-instruction tasks, respectively. The cost savings arise from reducing repeated LLM calls, which otherwise require inputting the entire conversation history each time. However, the performance drop may be attributed to the agent's tendency to greedily register multiple tools at once, which introduces complexity for each action taking. This increased complexity makes it easier for the agent to incorrectly select a tool from the larger pool, compared to having only a single registered tool, potentially leading to error propagation.

### 3.4.2 TOOL NAMES VS. DESCRIPTIONS INFORMATION IN TOOL REGISTRATION

We then investigate the feasibility of incorporating both tool names and descriptions in the tool registration process. We aim to address the following questions: Is the tool name sufficient for accurate tool registration? Does the inclusion of tool descriptions enhance registration performance? We also examine whether this modification affects the associated costs. To evaluate these questions, we conduct experiments using the G2-instruction and G3-instruction subsets, incorporating all available tool descriptions for registration within the ReAct framework, leveraging the arguments from `EcoAct` in the GPT-4o model. The results of our experiments are presented in Table 4.

From the results, we could observe that the including tool descriptions for tool registration does not necessarily lead to a noticeable improvement in performance. However, this approach incurs a

Table 4: We compared two variants of `EcoAct` for tool registration: (1) using tool names only, and (2) using both names and descriptions. Experiments on G2/G3-I subsets from ToolBench benchmark, using `EcoAct` to augment the ReAct algorithm on the GPT-4o model, showed that adding tool descriptions did not significantly improve performance but increased costs.

| Method | G2-instruction | | G3-instruction | | Average | |
|---|---|---|---|---|---|---|
| | PR (%) | Cost (¢) | PR (%) | Cost (¢) | PR (%) | Cost (¢) |
| ReAct | 20.5 | 6.7 | 18.4 | 10.7 | 19.5 | 8.7 |
| ReAct w/ `EcoAct` (Tool reg. by tool names) | 20.8 | **4.8** | **21.1** | **5.8** | **21.0** | **5.3** |
| ReAct w/ `EcoAct` (Tool reg. by tool names and des.) | **21.4** | 6.3 (↑ 31.3%) | 18.4 | 9.6 (↑ 65.5%) | 19.9 | 8.0 |

significant increase in cost, comparable to that of standard ReAct. Specifically, the cost increases 31.3% and 65.5% in G2-instruction and G3-instruction respectively. This finding suggests that tool names alone provide sufficient information for the agent to perform correct tool registration. This is because the context of tool descriptions is obviously larger than tool names. Consequently, the inclusion of tool descriptions may be unnecessary and could result in substantial cost increases.

# 4 RELATED WORKS

Large language models (LLMs) represent a major breakthrough in artificial intelligence, prompting an increasing body of research dedicated to employing LLMs in the construction of autonomous agents capable of performing complex tasks (Xi et al., 2023; Wu et al., 2023b; Peng et al., 2023; Shridhar et al., 2020; Song et al., 2024; Wu et al., 2024; Zhang et al., 2023; Ma et al., 2024). In these LLM-based agents, the ability to leverage external functions, tools, or actions to interact with the environment or solve sub-tasks is crucial. These external tools empower agents to go beyond natural language processing. For instance, LLM agents equipped with scientific tools can conduct scientific research (Bran et al., 2023; Ghafarollahi & Buehler, 2024), while those integrated with robotic systems can perform robotic manipulation tasks (Ahn et al., 2022; Huang et al., 2023).

To enable agents to use external tools, they must undergo a process called *tool registration*, where relevant tool information is integrated into the LLM's context prior to the agent taking action. This process becomes challenging when the number of available tools exceeds the context limits. One approach to mitigate this limitation is through retrieval-augmented generation (RAG) (Lewis et al., 2020; Gao et al., 2023). For example, Patil et al. (2023); Li et al. (2023) use a pre-trained text embedding model to retrieve relevant tools from a large tool pool. Similarly, Qin et al. (2023) trained an additional API retriever to identify essential tools using curated tool retrieval data.

To handle user queries with registered tools, various reasoning algorithms for LLM agents have been explored recently (Yao et al., 2022; Qin et al., 2023). Specifically, Yao et al. (2022) propose an approach that interleaves the generation of reasoning traces with tool-using actions, leading to more reliable and factual responses. Qin et al. (2023) introduce DFSDT, a decision tree-based method that expands the search space, increasing the likelihood of identifying a valid tool-using path. However, these reasoning algorithms do not integrate with the tool registration process, which can result in unnecessary costs due to the registration of irrelevant tools. In contrast, our approach seamlessly incorporates tool registration into these reasoning algorithms, allowing agents to autonomously reason about and register only the necessary tools, thereby avoiding such inefficiencies.

# 5 CONCLUSION

In this work, we propose `EcoAct`, a simple yet effective approach that seamlessly integrates tool registration into the intrinsic reasoning processes of LLM agents. The core concept involves initializing the agent with a meta-tool named *tool_register*, which enables the agent to selectively register tools deemed useful based on their names at each time step. This action allows the agent to avoid indiscriminately incorporating all candidate tools into its context, instead retaining only relevant information across reasoning steps, thereby achieving significant cost savings. We evaluate `EcoAct` on the ToolBench dataset, augmenting various reasoning methods, and demonstrate that `EcoAct` significantly reduces computational costs while maintaining comparable performance.

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

## A    REPRODUCIBILITY

In order to facilitate the peer review of the ICLR 2025 submission of our paper, we provide an anonymized link to our source code on: `https://shorturl.at/qqy21`.

## B    MORE DETAILS ABOUT TOOLBENCH

ToolBench is a large-scale tool-usage dataset comprising 16,464 real-world RESTful APIs across 49 categories from the RapidAPI Hub. All queries in this benchmark were generated by prompting ChatGPT to create diverse tasks involving these APIs, covering both single-tool and multi-tool usage scenarios. Through careful human evaluation, the authors determined that the generated instructions exhibit high diversity, reflecting a wide range of practical applications. This benchmark has been widely adopted as a standard evaluation tool in several studies (Du et al., 2024; Ye et al.).

### B.1    SUBSETS INFORMATION

The dataset is categorized into three levels: G1, G2, and G3, which correspond to single-tool instructions, intra-category multi-tool instructions, and intra-collection multi-tool instructions, respectively. Each level is further subdivided into three subcategories:

- Instruction: unseen instructions using the same set of tools as in the training data.
- Tool: unseen tools from previously encountered category as those in the training data.
- Category: unseen tools from an entirely different, previously unobserved category

However, since our `EcoAct` algorithm does not rely on training data, the distinctions between these three subsets are minimal.

### B.2    EVALUATION PROTOCOL

We adopt the same pass rate evaluation protocol as outlined in AnyTool (Du et al., 2024). In the original ToolBench benchmark, the authors employ a two-stage evaluation process. In the first stage, ToolBench uses an LLM (GPT-4 in our paper) to assess whether the selected API candidates can address the query, classifying them as either 'solvable' or 'non-solvable'. For queries deemed 'solvable', the LLM then evaluates the effectiveness of the solution, labeling it as either 'solved' or 'unsolved'. The pass rate is calculated using the following equation:

$$\text{Pass Rate} = \frac{\text{Non-solvable} + \text{Solved}}{\text{Non-solvable} + \text{Solved} + \text{Unsolved}} \quad (4)$$

A key issue with this evaluation protocol arises when there is a large number of 'non-solvable' queries identified by GPT-4. This can result in an artificially high pass rate, despite many queries remaining unsolved. To mitigate this, Du et al. (2024) conducted a manual review of all queries, retaining only those that can be resolved. Consequently, the pass rate is calculated using the following equation:

$$\text{Pass Rate} = \frac{\text{Solved}}{\text{Solved} + \text{Unsolved}} \quad (5)$$

More information of this evaluation protocol could be found in the original paper (Du et al., 2024). In terms of cost calculation, the monetary cost is computed based on the corresponding pricing from Microsoft Azure.

## C    MORE IMPLEMENTATION DETAILS

When using AnyTool to retrieve tools for each query, we set the maximum size of the API-Candidate Pool to 64, drawing on the findings of the AnyTool paper, which suggest that a pool size of 64 nearly

saturates performance. Additionally, we increased the maximum reasoning steps to 24, up from the default of 12, to explore the behavior of `EcoAct` under conditions without budget constraints.

# D PROMPTS

## D.1 PROMPT DESIGN FOR ECOACT

Table 5: Prompt for `EcoAct`.

```
You are AutoGPT, you can use many tools (functions) to do the
following task.  First I will give you the task description,
and your task start.
At each step, you need to give your thought to analyze the
status now and what to do next, with a function call to
actually excute your step.
After the call, you will get the call result, and you are now
in a new state.  Then you will analyze your status now, then
decide what to do next..  After many (Thought-call) pairs,
you finally perform the task, then you can give your finial
answer.
Remember:  1.the state change is irreversible, you can't go
back to one of the former state, if you want to restart the
task, say "I give up and restart".  2.All the thought is
short, at most in 5 sentence.  3.You can do more then one
trys, so if your plan is to continusly try some conditions,
you can do one of the conditions per try.
Let's Begin!
Task description:  You should use functions to help handle
the real time user querys.  But every function needs to be
selected using "function_selection" function before use it.
Remember:  1.ALWAYS call "Finish"function at the end of the
task.  And the final answer should contain enough information
to show to the user,If you can't handle the task, or you find
that function calls always fail(the function is not valid
now), use function Finish->give_up_and_restart.  2.  do not
call the function you have not successfully selected.
```

## D.2 TOOL_REGISTER

```
{
    "name": "function_register",
    "description": "I have given you a list of functions (names),
        please call this function to choose one of them that may
        be useful. The function you choose should be the one that
        you think is most useful in the current state. After you
        make function selection using this function, I will give
        you the detailed information of your selected function.
        You can then call the function you selected with
        appropriate inputs if you think the function is useful.",
    "parameters": {
        "type": "object",
        "properties": {
            "function_name": {
                "type": "string",
                "description": "the name of the function you want
                    to call",
            }
        }
```

```
702       },
703       "required": ["function_name"],
704   }
705
706
```

### D.3   PROMPT FOR REACT/DSFDT

Table 6: Prompt for Creating ReAct/DSFDT.

```
You are AutoGPT, you can use many tools (functions) to do the
following task.  First I will give you the task description,
and your task start.
At each step, you need to give your thought to analyze the
status now and what to do next, with a function call to
actually excute your step.
After the call, you will get the call result, and you are now
in a new state.  Then you will analyze your status now, then
decide what to do next..  After many (Thought-call) pairs,
you finally perform the task, then you can give your finial
answer.
Remember:  1.the state change is irreversible, you can't go
back to one of the former state, if you want to restart the
task, say "I give up and restart".  2.All the thought is
short, at most in 5 sentence.  3.You can do more then one
trys, so if your plan is to continusly try some conditions,
you can do one of the conditions per try.
Let's Begin!
Task description:  You should use functions to help handle
the real time user querys.  Remember:  1.ALWAYS call
"Finish"function at the end of the task.  And the final answer
should contain enough information to show to the user,If
you can't handle the task, or you find that function calls
always fail(the function is not valid now), use function
Finish->give_up_and_restart.  2.Do not use origin tool names,
use only subfunctions' names.  You have access of the
following tools:
```

### D.4   PROMPT FOR PASS RATE EVALUATIONS

#### D.4.1   PROMPT TEMPLATE FOR VERIFYING WHETHER THE QUERY HAS BEEN RESOLVED

```
----------------------------------------------------------------
<function>
<name>check_answer_status</name>
<description>
Giving the query and answer, you need give 'answer_status' of the
  answer by following rules:
1. If the answer is a sorry message or not a positive/straight
   response for the given query, return "Unsolved".
2. If the answer is a positive/straight response for the given
   query, you have to further check.
2.1 If the answer is not sufficient to determine whether the solve
    the query or not, return "Unsure".
2.2 If you are confident that the answer is sufficient to
    determine whether the solve the query or not, return "Solved"
    or "Unsolved".

Query:
{query}
```

```
Answer:
{answer}

Now give your reason in "content" and `answer_status` of JSON to `
    check_answer_status`.
</description>
</function>
------------------------------------------------------------------
<function>
<name>parse_answer_status</name>
<description>
Giving the query and the correspond execution detail of an answer,
     you need give `answer_status` of the answer by following
    rules:
1. If all 'tool' nodes' message indicate that there are errors
    happened, return "Unsolved"
2. If you find the information in the "final_answer" is not true/
    valid according to the messages in 'tool' nodes, return "
    Unsolved"
3. If you are unable to verify the authenticity and validity of
    the information, return "Unsure"
4. If there are 'tool' node in the chain contains successful func
    calling and those calling indeed solve the query, return "
    Solved"

Query:
{query}
Answer:
{answer}

Now you are requested to give reason in "content" and `
    answer_status` of JSON to `parse_answer_status`.
</description>
</function>
------------------------------------------------------------------
```

### D.4.2 Prompt template for verifying whether the query is solvable

```
------------------------------------------------------------------
<function>
<name>check_task_solvable</name>
<description>
Please check whether the given task solvable with following rules:
1. If the `query` provide invalid information (e.g. invalid email
    address or phone number), return "Unsolvable"
2. If the `query` needs more information to solve (e.g. the target
     restaurant name in a navigation task), return "Unsolvable"
3. If you are unable to draw a conclusion, return "Unsure"
4. If the currently `available_tools` are enough to solve the
    query, return "Solvable"

Task:
{task}

Now give your reason in "content" and `task_status` of JSON to `
    check_task_solvable`.
</description>
</function>
------------------------------------------------------------------
```

