# OpenReview forum: "EcoAct: Economic Agent Determines When to Register What Action"
_ICLR.cc/2025/Conference — Submitted to ICLR 2025_

### Official Review · Reviewer_VgpE · 2024-10-31

**Soundness:** 3
**Presentation:** 3
**Contribution:** 3
**Rating:** 5
**Confidence:** 4

**Summary:**

The paper introduces EcoAct, an algorithm that optimizes the tool registration process for Large Language Models (LLMs) acting as agents. Traditionally, LLMs incorporate all candidate tools into their context, leading to inefficiencies. EcoAct allows LLMs to selectively register tools as needed, reducing computational costs by over 50% in multi-step reasoning tasks without sacrificing performance. It integrates tool registration into the reasoning process, making it universally applicable to LLM agents. Extensive experiments on the ToolBench benchmark demonstrate significant cost savings and maintained performance across various models and reasoning methods.

**Strengths:**

* Clear writing and good presentation.
* EcoAct is effective and reduce token cost without performance drop.

**Weaknesses:**

* In Figure 3a, why gpt4-turbo cost using EcoAct is higher (level 1)? Please explain this.
* Lack of technical novelty. It seems like that EcoAct only a few changes compared to React (from Tabel 5 and 6).

**Questions:**

Please see weakness

---

> ### Author Response · Authors · 2024-11-23
> **Official Response**
>
> **[1. Why gpt4-turbo cost using EcoAct is higher (level 1)? Please explain this.]**
>
> Thanks for the suggestion. We illustrate it in section 3.2. “This occurs because EcoAct introduces an additional LLM calling procedure for tool registration step. When the number of tools for specific queries is small, the cost of incorporating all tool information may be less than the cost of this extra LLM call, causing the advantages of our method to diminish.” Please note that our method is designed for large-scale tool use. In cases where only a limited number of tools are available, users may opt to use alternative algorithms, such as ReAct.
>
> **[2. Lack of technical novelty.]**
>
> We respectfully argue that simplicity should not be considered a disadvantage of a method; rather, it is an advantage. The simplicity of EcoAct enables it to function as a plug-and-play technology for large-scale tool usage, as demonstrated in the experimental section. This will lead to much broader impact across a wide range agent frameworks in the real world.

---

> ### Comment · Reviewer_VgpE · 2024-11-25
>
> Thank you for responding and answering my questions. However, I think the novelty of this paper is still limited. Compared to ReAct, EcoAct just adds a tool registration and has a few changes compared with previous work. The technical contribution seems incremental. I decide to keep my score.

---

### Official Review · Reviewer_b4WG · 2024-10-31

**Soundness:** 3
**Presentation:** 3
**Contribution:** 2
**Rating:** 5
**Confidence:** 3

**Summary:**

The paper presents a variant of the ReAct framework by introducing a tool registration process. The rationale is that this would help the agent not having to carry the whole tool registering through the steps and could reduce the computational cost. The paper shows some good experimental results which demonstrate comparable performance and reduced computational cost compared to the ReAct baseline. The step introduced is simple and generalisable for other multi-tool LLM algorithm/framework.

**Strengths:**

The paper is well-written and has a clear structure. The rationale of the framework and the experiments have been presented in a logical and detailed way. The paper provided both a conceptual description of the problem and practical examples to illustrate how EcoAct could help improve the sequence of thought-action while reducing computational cost. The step proposed is simple and generalisable for most single and multi LLM frameworks. The experimental results show a good potential and support the hypothesis that this modified framework does not affect the performance of ReAct and can improve on computational cost. Overall, the paper demonstrates good quality and clarity.

**Weaknesses:**

The paper is generally well written and presented but might further improve if it demonstrates more of the following:
* Presentation: Minor typo: "results" - line 386
* Experimental design and data analysis:
    * While the experiments were conducted on ToolBench and captured a variety of tools and subsets (G1, G2 and G3), apart from the general stats of the pass rate, the analysis did not offer a lot of insights on how these subsets differ and why/how the tool selection step of EcoAct helps in each of these subsets. There is some hypothesis proposed in the Overall result section but the paper did not delve further into any analysis to support or provide more evidence on the " method’s ability to address the ”needle-in-a-haystack” problem" (line 317).
    * The results have been largely based on GPT-4o and GPT-4-turbo. The results might be more convincing and comprehensive if there are more baseline for comparison, such as using other LLM models.
* While the experimental results show good cost improvement compared to the baseline of ReAct, it might need to distinguish the novelty and its computational advantage to other related works that also improve on the efficiency of API calling/token cost of tool information, such as:
    * Wu, Mengsong, Tong Zhu, Han Han, Chuanyuan Tan, Xiang Zhang, and Wenliang Chen. "Seal-Tools: Self-Instruct Tool Learning Dataset for Agent Tuning and Detailed Benchmark." arXiv preprint arXiv:2405.08355 (2024).
    * Wang, Pei, Yanan Wu, Zekun Wang, Jiaheng Liu, Xiaoshuai Song, Zhongyuan Peng, Ken Deng et al. "MTU-Bench: A Multi-granularity Tool-Use Benchmark for Large Language Models." arXiv preprint arXiv:2410.11710 (2024).
    * Wang, Renxi, Xudong Han, Lei Ji, Shu Wang, Timothy Baldwin, and Haonan Li. "ToolGen: Unified Tool Retrieval and Calling via Generation." arXiv preprint arXiv:2410.03439 (2024).

**Questions:**

* The paper clearly presents and demonstrates that there is a reduction in computational cost by letting the LLM choose to register a tool rather than being provided with the whole toolset. However, could you clarify whether the tool registration process can be a bottleneck if the LLM needs to iteratively select/register a tool (or a sequential set of tools) in order to achieve the task?
* Also, is there any extra time/steps incurred for registering/ querying the tool?
* Could you also clarify your results in Section 3.4.1 please? Why do using multiple tool registration significantly reduces the performance  vs the React basecase and the EcoAct single tool case? Why your explanation refers to the tendency to registering multiple tools leading to increased complexity, how is it differ to the React basecase when there are multiple tools also being registered?

**Details Of Ethics Concerns:**

This is a conceptual framework of tool selection which has no direct ethical concerns in its current form.

---

> ### Author Response · Authors · 2024-11-23
> **Official Response**
>
> Thanks for the suggestions. Please find our response to your comments below.
>
> **[1. Lack  details of different  subsets in ToolBench.]**
>
> We explain the differences between the various subsets at the beginning of Section 3.1 and will ensure to emphasize this further in future versions.
>
> **[2. Including more models & Including more related works.]**
>
> Thanks. We plan to include additional experiments with open-source models and the suggested related works to discuss in our final version.
>
> **[3. Whether the tool registration process can be a bottleneck if the LLM needs to iteratively select/register a tool (or a sequential set of tools) in order to achieve the task?]**
>
> All of our experiments are conducted on queries that involve a sequential set of tools. As demonstrated in the experimental section, the tool registration process does not present a bottleneck.
>
> **[4. Is there any extra time/steps incurred for registering/querying the tool?]**
>
> Yes, it does. As shown in the experimental section, it does not lead to an increase in cost, but instead results in a cost savings of over 50%.
>
>  **[5. Clarifying the results in Section 3.4.1.]**
>
> Thank you for your suggestions. The key difference between using the original ReAct and EcoAct with multiple tool registrations lies in the inclusion of tool registration contexts/prompts in the EcoAct history, while ReAct is focused solely on tool usage. This added complexity makes it more likely for the agent to perform incorrect actions compared to ReAct, which is dedicated solely to tool usage.

---

> > ### Comment · Reviewer_b4WG · 2024-11-25
> >
> > Thank you for responding and answering my questions. I appreciate your effort in providing extra clarification informations. However, I will need further concrete results and analysis in order to change my score. For now, I am inclined to retain the current score. Specifically, I'd be interested to see the following results and analysis:
> > 1. Analysis with regard to the different nature of subsets in ToolBench: this is not about the description of the subsets but about analysing how the methods perform across the subset and why/how the tool selection step of EcoAct helps in each of these subsets. This could reveal whether your methods work across the subsets or a specific set of problems/datasets.
> > 2. Including more models: need actual experiment results with other open-source models

---

### Official Review · Reviewer_WLfz · 2024-11-02

**Soundness:** 3
**Presentation:** 3
**Contribution:** 2
**Rating:** 5
**Confidence:** 4

**Summary:**

The paper proposes a general method for reducing token usage in LLM agent workflows and assesses its effects on performance and cost. The method involves withholding from the LLM by default the details of the available tools. Instead, apart from the historical context of the task at hand, the LLM receives, at each step, only the names of the available tools, the details of a "tool registering" tool, and the details of previously registered tools (initially, none). Use of a tool must be preceded by a call to the "tool registering" tool in order to register the desired tool, after which the tool becomes available for use. This is a general method, and can be applied to most LLM agent workflows. The experiments are done in the context of ToolBench, and show a substantial decrease in cost with a small effect on performance. The paper also discusses two alternative setups: one where the LLM agent is able to register multiple tools at once (which performs substantially worse than either the vanilla or the single-tool registering setups), and another where the LLM agent also has access to all tools' descriptions, rather than just their names (which does not offer as much reduction in cost).

**Strengths:**

**Originality**

The method introduced by the paper seems original. There exists some work on dynamically managing an agent's tool library, but I'm not aware of this particular approach having been investigated before (this is a bit surprising to me, as the approach is relatively simple).


**Quality**

The paper's experiments are well-motivated and follow naturally from the research question. The ablation experiments are also well-chosen and add flavor to the results. There do not seem to be any methodological issues, and the results clearly support the claims. I don't find any of the empirical claims to be dubious.


**Clarity**
The paper is clearly-written and all its aspects are well-motivated. The metrics and figures are well-chosen and clearly support the claims. The question, method, experiments, and results all follow clearly from each other. When appropriate, fruitful discussions about the mechanisms at play are discussed and add clarity to the text (e.g. section 3.3.2 is interesting and substantially enhanced my understanding of the results).


**Significance**

Agentic workflows with tools are being increasingly adopted, and especially as inference-heavy methods gain more prominence, reducing token usage with no detriment to performance may become increasingly important. That said, I don't think the exact problem this paper focuses is particularly significant (see Weaknesses).

**Weaknesses:**

1. The paper's experiments are done in the context of ToolBench, where very many tools may be available. This does not seem like a representative or particularly important setting to me. A majority of agentic workflows either take place on more constrained contexts, or use more general tools. In either case, the number of tools available to the agent is much smaller than in the context of the experiments. The authors acknowledge this and helpfully make the distinction in Figure 3, but it remains a significant limitation of the paper.

2. The paper's method is not compared with alternative simpler solutions. An example of a possible alternative would be to begin by asking the LLM to filter the list of available tools given the initial task, and then have it attempt to complete the task with only the selected tools. More complex combinations are possible (e.g. start with a list of filtered tools, but still allow the agent to register new tools). It would be very informative to see how the proposed method compares to such baselines.

3. The previous concern also exists at a more fundamental level: it can be argued that the proper solution for the problem the paper tackles is to make use of prompt caching. The context including the details of all available tools is constant across completion requests, and could be precomputed and reused across calls for very cheap. At the moment of writing this, both the OpenAI and Anthropic APIs support some form of prompt caching. It seems plausible that support for this will become even more widespread and more efficient, so the economic formulation of this work is not very compelling. There is still a surprising result here along the lines of the models being able to exercise good judgement about which tools they need at which times. I was surprised to see that the effect on performance was small. I think a framing of this problem and investigation in terms of the abilities of the language models is not too far away and is possibly very compelling.

**Questions:**

1. How much would you agree with my claim that the context of the experiments (containing many possible tools) is not very representative? What do you expect the results to look like in different settings, and why?

2. The URL provided in appendix A is not working for me (it times out with error 522). Would it be possible to confirm that it is working, or fix it if not? I'm especially interested in better understanding the tasks that were used for the evaluation. My current understanding from looking online is that they are mostly pretty simple, with the main challenge being deciding which tool is appropriate, and then making the correct call. If the difficulty of the tasks essentially reduces to choosing the right tool, then I'm less impressed by the results. On the other hand, if the tasks often require multiple steps that build upon each other and the agent must react to the outputs of the tools in nuanced ways, then I'm more impressed by the finding that the agent is fairly good at deciding which tools to use over the course of the task.

3. Did you test alternative methods for reducing the token usage in the agentic workflows in question? How confident are you that simple alternatives cannot outperform your method?

4. This isn't of much consequence, but the claim in section 2.1 that "the size of the tool space is equivalent to the size of the action space" seems false to me. Isn't it the case that many tools can be called in potentially infinite ways (via differences in the parameters), such that the action space is infinite while the tool space is finite?

---

> ### Author Response · Authors · 2024-11-23
> **Official Response**
>
> Thank you for your thoughtful and constructive comments. You indeed give us a lot of high-quality comments. Below is our response to your feedback.
>
> **[1. The paper's experiments are done in the context of ToolBench, where very many tools may be available. This does not seem like a representative or particularly important setting to me & Q1.]**
>
> We acknowledge that EcoAct may not yield significant cost savings when the number of equipped tools is relatively small (fewer than 10), as demonstrated in Section 3.2. However, we would like to emphasize that the primary scenarios we target involve situations where a large number of tools are available, rather than those with a small toolset. In cases where only a limited number of tools are available, users may opt to use alternative algorithms, such as ReAct. Additionally, would you kindly provide more details or examples of the settings you believe would be more representative? This would help us better address your concern and strengthen our work. Thanks!
>
> **[2. The paper's method is not compared with alternative simpler solutions & Q3.]**
>
> Thank you for your valuable suggestion.
>
> Baseline 1 - Let agent filter out tools beforehand:  We believe this native method may be limited, as it requires the agent to pre-determine which tools to use, rather than dynamically registering them based on actual usage. Essentially, EcoAct enables the agent to register alternative tools when some tools fail, whereas this native baseline does not provide that flexibility.
>
> Baseline 2 - Start with a list of filtered tools, but still allow the agent to register new tools: This baseline could be regarded as a combination of EcoAct with a native RAG method - agent as tool filter.
>
> Your suggestion is insightful, and we plan to incorporate it in our future version.
>
>
> **[3. Prompt caching may solve the same problems.]**
>
> We respectfully disagree with the suggestion that prompt caching could address the same issue. The reason for this is that tool information appears not only in the initial prompt, but also in each subsequent round of LLM interactions. While prompt caching can benefit the first round by enabling cache hits, it does not apply to intermediate stages of the process.
>
> **[4."the size of the tool space is equivalent to the size of the action space" seems false to me. ]**
>
> We totally agree with this and will correct these claims in future versions.

---

> > ### Comment · Reviewer_WLfz · 2024-11-26
> >
> > I thank the authors for the helpful responses.
> >
> > Overall, my decision recommendation remains unchanged at a "marginal reject".
> >
> > I find the answer to Q1 reasonable, but ultimately it still seems to me that the context in which the results are presented is a bit artificial, with ToolBench being a bit too extreme in the number of tools available.
> >
> > On the question of prompt caching, I think I fail to see the authors' point and I would still maintain that prompt caching could be an effective alternative. The current implementation of prompt caching in the Anthropic API, for example, does allow for caching the tool definitions (see [what can be cached](https://docs.anthropic.com/en/docs/build-with-claude/prompt-caching#what-can-be-cached)). So in this case, the full tool list with all definitions can be passed in the beginning, cached, and thereafter only be accessed via cache hits at every agent step. But regardless of the specifics of current API implementations, my point is that there is no technical barrier for this in general, and that caching as a technique in the abstract comes close to solving the problem in the abstract.

---

### Official Review · Reviewer_ioq7 · 2024-11-02

**Soundness:** 3
**Presentation:** 3
**Contribution:** 3
**Rating:** 5
**Confidence:** 3

**Summary:**

The paper introduces EcoAct, an innovative tool-using algorithm designed to enhance the efficiency of large language model agents in multi-step reasoning tasks. Traditional approaches integrate all candidate tools into the LLM’s context, making the process inefficient, as the agent must process unnecessary information. EcoAct addresses this by enabling the agent to selectively register tools as needed during each reasoning step. This method optimizes context length and reduces token usage, significantly lowering computational costs—achieving over 50% cost reduction in some tests—without compromising performance.

**Strengths:**

+ The paper is well written, with ideas presented clearly and in an organized manner.
+ The paper introduces a simple yet effective idea for saving tokens when using tools or solving problems, which could have practical implications for reducing costs.
+ The authors provide a thorough ablation study, offering valuable insights into the effects of different components (like Single-Tool vs. Multiple-Tool Registration) on model performance.

**Weaknesses:**

Model Selection and Analysis

- The evaluation focuses on two relatively strong models, but no open-source models are included, despite files named “llama*” and “davinci*” appearing in the provided code. This raises the question of why these models were not evaluated or discussed in the paper. Additionally, there is no analysis of model strengths or the tools used. For instance, an examination of how precisely tool descriptions and the tool register need to be defined for optimal cost and accuracy would be valuable. Specifically, exploring the threshold between model size/abilities and the level of detail in tool descriptions could offer insights into balancing model performance with cost.

Performance Issues and Explanation

- It’s concerning that models with descriptions tend to perform worse on average in G2 and G3 - Instruction (as shown in Table 4), yet no explanation is provided for this behavior.

- The effect of function names on model performance is not addressed. For example, what happens if function names are unintuitive (e.g., def do_or_not(*)) or if there are several similarly named functions (like “calculator” vs. “scientific calculator”)? This ambiguity could impact the model’s accuracy and effectiveness.

Code Quality and Documentation

- The provided code is poorly written, which could make understanding and reproducing the results challenging.

Visual and Presentation Inconsistencies

- The scale in Figure 3 (a) and (b) is inconsistent. Using a common scale across both graphs would allow for a clearer comparison of results.

Missing Evaluation Protocols

- The paper lacks details on the evaluation protocol, such as model temperature settings, maximum tokens, and other relevant parameters.

Including these analyses would add significant value to the paper, and I would be glad to raise my score if these areas are addressed.

**Questions:**

Q1. Why were no open-source models included in the evaluation, especially given that files named "llama*" and "davinci*" appear in the provided code? Could these models be tested or discussed?

Q2. Is there an analysis of the strengths of the models used and the tools provided? For example, how precisely do tool descriptions and the tool register need to be defined for optimal cost and accuracy?

Q3. Why do models with descriptions perform worse on average in G2 and G3 - Instruction (as shown in Table 4)? Could the authors provide an explanation for this behavior?

---

> ### Author Response · Authors · 2024-11-23
> **Official Response**
>
> Thank you for your constructive suggestion. Please find our response to your comments below.
>
> **[1. No open-source models are included &  Exploring the threshold between model size/abilities and the level of detail in tool descriptions could offer insights into balancing model performance with cost. & Q1]**
>
> Thanks for the suggestions. We plan to include additional experiments in future versions, incorporating a broader range of models, including open-source models of varying sizes.
>
> **[2. An examination of how precisely tool descriptions and the tool register need to be defined for optimal cost and accuracy would be valuable & Q2]**
>
> We would like to clarify that we do not have control over the tool descriptions to optimize cost and accuracy, as these are determined by the users and the specific application scenarios. In Section 3.4.2, we present conclusions regarding tool descriptions, indicating that, in certain scenarios (e.g., ToolBench), tool names alone are sufficient for tool selection.
>
> **[3. It’s concerning that models with descriptions tend to perform worse on average in G2 and G3 - Instruction (as shown in Table 4). & Q3]**
>
> Thanks for your suggestions. We also observed these interesting results. In terms of performance, we found that ReAct with EcoAct (tool registration by tool names) outperformed ReAct with EcoAct (tool registration by both tool names and descriptions), which in turn outperformed ReAct. We hypothesize that this may be due to the challenges faced by LLM agents when performing actions within a large context. Specifically, after retrieving tools in the ToolBench benchmark, the number of tools associated with each query is often quite large (typically more than 15), resulting in an expanded context for the retrieved tools. With more detailed information about these tools, the LLM may struggle to identify and select the correct tools, leading to a performance drop. This suggests that tool names alone provide sufficient information for tool selection, and adding descriptions may introduce unnecessary complexity in larger contexts.
>
> **[4. What happens if function names are unintuitive & Q2]**
>
> Indeed, the way tools are named can impact the tool registration process. However, we would like to clarify that tool registration in EcoAct is not a one-time procedure. If an agent registers an incorrect tool due to an uninformative tool name, it can quickly become aware of the issue by accessing the tool's detailed description and subsequently explore alternative tools. We will ensure this is made clearer in a future version of the paper. Thank you for your valuable suggestion.
>
> **[5. Implementation details & Documentations.]**
>
> Thanks for these constructive suggestions. These suggestions are indeed valuable to us. We will definitely revise our work according to these comments.

---

> > ### Comment · Reviewer_ioq7 · 2024-11-25
> >
> > Dear Authors,
> >
> > Thank you for your detailed response. While I appreciate your plan to include additional experiments in future versions, I cannot raise the score based on promises of future work - I need to see concrete experimental results in the current submission.
> >
> > Specifically, I would be very interested in seeing a plot or analysis that examines the relationship between tool description length and model performance across different model sizes. This could reveal how smaller models interact with EcoAct and whether they require more detailed tool descriptions compared to larger models.

---

### Official Review · Reviewer_RQft · 2024-11-03

**Soundness:** 2
**Presentation:** 3
**Contribution:** 2
**Rating:** 3
**Confidence:** 4

**Summary:**

This paper introduces EcoAct, a novel approach designed to enhance tool registration efficiency for Large Language Model (LLM) agents by selectively registering tools during reasoning tasks. Unlike current methods, which pre-register all candidate tools regardless of their necessity, EcoAct integrates tool registration into the reasoning process, optimizing context length and computational costs. The authors demonstrate through extensive experiments that EcoAct reduces computational costs by over 50% while maintaining performance on complex, multi-step reasoning tasks.

**Strengths:**

**Explicit Tool Registration Step:** EcoAct introduces an explicit tool registration step early in the reasoning process to select the most appropriate tool, effectively reducing token costs. This proactive approach helps optimize computational efficiency by only registering tools when they are needed, rather than pre-registering all possible tools.

**Comprehensive Experimental Results:** The paper presents comprehensive experiments and results that clearly demonstrate the effectiveness of the proposed method. The evaluations are easy to follow and provide strong empirical evidence for EcoAct's ability to reduce computational costs while maintaining performance, making it easier for readers to understand the benefits of the approach.

**Clear Presentation and Writing:** The paper is well-written, with a clear and logical flow that makes it easy to understand the motivations, methods, and results. The clarity in presenting complex concepts and experimental findings reflects careful attention to detail and greatly enhances the accessibility of the work, making it approachable for a broad audience.

**Weaknesses:**

**Need for Comparative Experiments Against RAG-Based Methods:** The paper does not clearly demonstrate how EcoAct performs in comparison to Retrieval-Augmented Generation (RAG)-based tool selection methods. In RAG-based approaches, relevant tool candidates are retrieved by computing similarity measures—such as cosine similarity between semantic embeddings—between the user's query and tool representations stored in a vectorized retrieval index. While the paper provides a solid experimental evaluation, it is crucial for the authors to present comparative experiments against RAG-based methods. Such methods could potentially reduce token usage, thereby achieving benefits similar to EcoAct in terms of efficiency and cost savings. A direct comparison would help to highlight the unique advantages of EcoAct, as well as any areas where it may fall short compared to RAG-based approaches.

**Dependence on Tool Naming for Effective Selection:** EcoAct's performance is heavily influenced by how tools are named. The approach assumes that tool names are descriptive enough for effective selection, but this assumption may not always hold true. The authors should specify any requirements or conditions for tool names to ensure proper functioning of EcoAct. Additionally, experiments examining different levels of descriptiveness or ambiguity in tool names would help clarify the robustness of EcoAct under varying conditions.

**Lack of Novelty and Details:** Even though the effectiveness of the proposed method in reducing token costs is evident, the approach of simply adding a tool registration step is too simplistic. The authors have not considered various conditions for ensuring the robustness of EcoAct.

**Lack of Qualitative Analysis of Tool Registrations:** The paper lacks a qualitative analysis of the tool registration process, including notable examples of when and why specific tools are registered. Providing examples would help illustrate the types of tools that are beneficial for registration and shed light on EcoAct's decision-making process, making it easier for readers to understand the practical implications and real-world applicability of the method.

**Questions:**

1. Have you considered other approaches for minimizing tool registration overhead besides selective registration, e.g., RAG, fine-tuning, etc., for tool selection?

2. How would EcoAct recover from errors when initial tool selections fail, especially under ambiguities present in the tool names?

3. How would EcoAct perform if there are multiple tools that are feasible for the user query?

4. Why didn’t you consider employing open source LLMs for experiments?

5. Are there any comparisons with related works like ToolLlama, Gorilla, etc.?

---

> ### Author Response · Authors · 2024-11-23
> **Official Response**
>
> We sincerely thank the reviewer for the insightful comments. Please find our response to your comments below.
>
> **[1. Lack of comparisons with RAG methods.]**
>
> Thank you for your constructive suggestions. EcoAct is fundamentally a reasoning method that is agnostic to specific RAG (Retrieval-Augmented Generation) methods, the prototype is (1) RAG methods are used to retrieve tools, and (2) EcoAct is then applied to use those tools efficiently. For instance, in the experimental section, we employ the state-of-the-art tool retrieval method, AnyTool, to retrieve tools for each query in all the experiments presented in this paper and then perform EcoAct.
>
> We also recognize that including RAG as a direct baseline would help better demonstrate the advantages of our method. Thus, the experimental design could be structured as follows: (1) RAG, (2) RAG + EcoAct, (3) EcoAct. We will integrate these great suggestions in our final version. Thanks for the suggestions.
>
>
> **[2.  EcoAct's performance is heavily influenced by how tools are named. & How would EcoAct recover from errors when initial tool selections fail, especially under ambiguities present in the tool names?]**
>
> Indeed, the way tools are named can impact the tool registration process. However, we would like to clarify that tool registration in EcoAct is not a one-time procedure. If an agent registers an incorrect tool due to an uninformative tool name, it can quickly become aware of the issue by accessing the tool's detailed description and subsequently explore alternative tools. We will ensure this is made clearer in a future version of the paper. Thank you for your valuable suggestion.
>
>
> **[3. The approach of simply adding a tool registration step is too simplistic.]**
>
> We respectfully argue that simplicity should not be considered a disadvantage of a method; rather, it is an advantage. The simplicity of EcoAct enables it to function as a plug-and-play technology for large-scale tool usage, as demonstrated in the experimental section (Section 3), which will lead to a much broader impact across agent frameworks in the real world.
>
> **[4. Lack of case-study.]**
>
> Thanks for the suggestion. We will include case study in our final version.
>
> **[5. Have you considered other approaches for minimizing tool registration overhead besides selective registration? & Are there any comparisons with related works like ToolLlama, Gorilla, etc.?]**
>
> Our method is primarily designed as a reasoning framework for large-scale tool usage, which can be integrated with other efficient tool utilization technologies (e.g., fine-tuning, RAG, etc.). We will clarify this further in final versions of the paper.
>
> **[6. How would EcoAct perform if there are multiple tools that are feasible for the user query?]**
>
> As demonstrated in Section 3.3.2 and illustrated in Figure 1, the agent employs its intrinsic reasoning mechanism to perform multiple tool selections, thereby evolving its skill set.
>
> **[7. Why didn’t you consider employing open source LLMs for experiments?]**
>
> Thanks for the suggestion. We will add more experiments including open-source models in our final version.

---

> > ### Comment · Reviewer_RQft · 2024-11-25
> >
> > Thank you for providing detailed responses to my comments. I appreciate your consideration of these suggestions for improving the paper. I look forward to reviewing the final version to evaluate how effectively the revisions address the identified weaknesses.

---

### Official Review · Reviewer_3scp · 2024-11-05

**Soundness:** 1
**Presentation:** 2
**Contribution:** 1
**Rating:** 3
**Confidence:** 4

**Summary:**

This paper proposes a cost-reduction method in which tools are registered only when required by the LLM, thereby optimizing context usage. The method is claimed to maintain performance while reducing costs with models like GPT-4 and GPT-4-turbo.

**Strengths:**

The authors conduct experiments using ReAct and DFSDT on the ToolBench benchmark, demonstrating cost reductions.

**Weaknesses:**

The proposed method reduces context length at each reasoning step by listing all tools at the beginning of the instruction, then registering only a single tool as needed. While this approach yields cost savings, the benefit is somewhat straightforward, and the performance impact is minimal. The method primarily benefits prompting techniques that provide all tool information at each step. Additionally, the cost reductions are specific to API expenses, making it unclear how this approach would apply to open-source models that run locally with comparable performance. This substantially limits the method's applicability and overall impact.

**Questions:**

1. What is the maximum number of steps the LLM can perform before failure is declared?

---

> ### Author Response · Authors · 2024-11-23
> **Official Response**
>
> Please find our response to your comments below.
>
> **[1. The performance impact is minimal.]**
>
> Our approach is primarily focused on optimizing computational costs rather than improving performance. We do not claim in this paper that EcoAct is intended to enhance performance. As shown in the experimental section, EcoAct functions as a plug-and-play component that reduces tool usage costs while maintaining comparable performance to its base reasoning method.
>
> **[2. The cost reductions are specific to API expenses.]**
>
> Yes, this is precisely the motivation behind our research, as discussed in Section 2.2. We designed EcoAct specifically to address scenarios with high API costs. This is not a limitation of our method.
>
>
> **[3. The method primarily benefits prompting techniques that provide all tool information at each step.]**
>
> We would like to clarify the following points:
> We do not provide complete tool information at every step; rather, we offer partial information to guide the tool selection process.
> Our method does incorporate prompting techniques in certain procedures, and we do not consider using prompting as the drawback of the approach.
>
>
> **[4. What is the maximum number of steps the LLM can perform before failure is declared?]**
>
> We show the number in Appendix. C. It is set to 24, up from the default of 12 in ToolBench, to explore the behavior of EcoAct under conditions without budget constraints.

---

### Meta-Review · Area_Chair_wbnq · 2024-12-18

**Metareview:**

This paper studies an emerging setting in using large language models (LLMs) by allowing LLMs to function as agents that can make use of external tools in solving some tasks. It proposes a method that allows it to focus on only some of the candidate tools that are more likely to be useful and hence makes it more efficient than considering all candidates.

Major strengths:
- Having an efficient and effective way to select the right external tools to use is essential to the agentic approach to using LLMs. The paper studies an important problem in this emerging direction.
- Comprehensive experiments and ablation studies are presented.

Major weaknesses:
- The improvement brought by the proposed method may not be significant when the number of tools is small.
- The incorporation of RAG-based methods is not considered in the paper to demonstrate better how the proposed method can help.
- The effectiveness of the proposed method depends on such things as naming and detailed descriptions of the tools.
- The LLMs included are limited.

Despite its merits, some important issues need to be addressed to make the paper more ready for publication. The authors are encouraged to improve their paper for future submission by considering the comments and suggestions of the reviewers.

**Additional Comments On Reviewer Discussion:**

Most reviewers participated in the discussions. The authors promised some future extensions in response to the comments from some reviewers. However, the reviewers would like to see concrete results, instead of promises, before changing their ratings and recommendations.

---

### Decision · Program_Chairs · 2025-01-22

Reject